# STABLE VIDEO-DRIVEN PORTRAITS

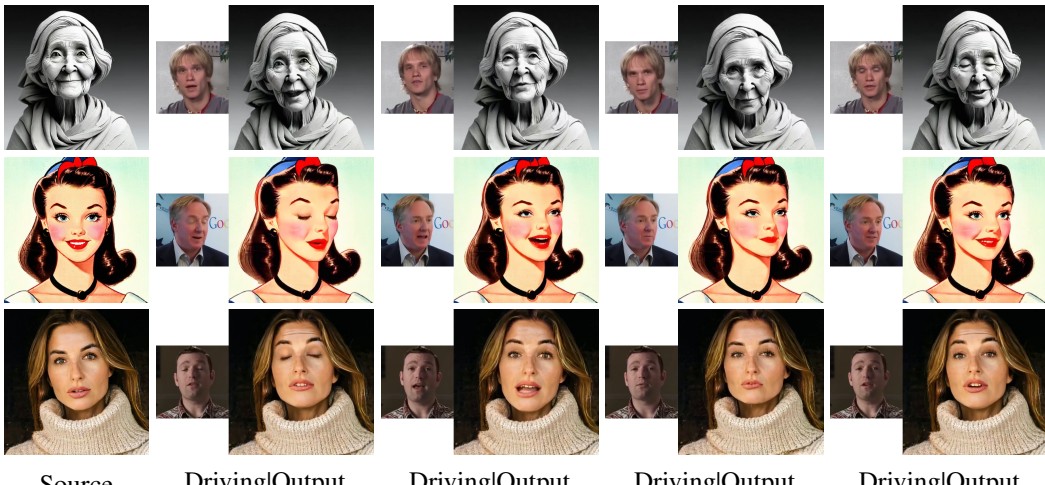

Figure 1: Our model takes a single source image and a driving video as input and can synthesize a high-quality video of the source image following the expressions and poses of the driving video. Although our model is trained on real human videos, it generalizes well to stylized human-like images as well.

## ABSTRACT

Portrait animation aims to generate photo-realistic videos from a single source image by reenacting the expression and pose from a driving video. While early methods relied on 3D morphable models or feature warping techniques, they often suffered from limited expressivity, temporal inconsistency, and poor generalization to unseen identities or large pose variations. Recent advances using diffusion models have demonstrated improved quality but remain constrained by weak control signals and architectural limitations. In this work, we propose a novel diffusion-based framework that leverages masked facial regions—specifically the eyes, nose, and mouth—from the driving video as strong motion control cues. To enable robust training without appearance leakage, we adopt cross-identity supervision. To leverage the strong prior from the pre-trained diffusion model, our novel architecture introduces minimal new parameters that converge faster and help in better generalization. We introduce spatial-temporal attention mechanisms that allow inter-frame and intra-frame interactions, effectively capturing subtle motions and reducing temporal artifacts. Our model uses history frames to ensure continuity across segments. At inference, we propose a novel signal fusion strategy that balances motion fidelity with identity preservation. Our approach achieves superior temporal consistency and accurate expression control, enabling high-quality, controllable portrait animation suitable for real-world applications.

## 1 INTRODUCTION

Portrait animation aims to synthesize photo-realistic changes in facial expressions and head poses. This has a wide range of applications in telepresence, virtual reality, gaming, and entertainment. High-fidelity systems often rely on expensive multiview video capture setups to acquire large volumes

of data for realistic synthesis. More recent approaches achieve improved synthesis quality using only a single image at test time. While this setup enables broader accessibility and deployment, it introduces challenges in generalizing to unseen identities and expressions. There are also various ways to control expressions and head poses. Some methods provide explicit control using audio, facial landmarks, emotions, or a video of another person as driving input. Among these, video-driven methods offer the highest degree of control and are especially desirable in many practical scenarios. In this work, we address the task of generating a video of a source identity from a single image, such that it faithfully reenacts the motion and expressions from a driving video. (See Fig. 1).

Earlier works addressed this task by using pretrained 3D morphable models to explicitly disentangle identity and expression. However, due to their limited expressiveness and the inherent ambiguity in representing identity and motion separately, these methods often produce results that lack realism. Subsequent approaches introduced warping-based techniques that disentangle appearance from motion, using feature-space warping and neural rendering for output synthesis. While these methods improved visual quality, they generally lacked temporal modeling, crucial for capturing subtle motion, such as during speech. Additionally, they struggled with generalizing to large pose variations and often produced artifacts in regions like the shoulders due to ambiguities in the warping fields.

Recently, diffusion models have revolutionized image and video synthesis tasks. They provide rich prior knowledge and latent spaces that can be easily fine-tuned to use various control signals to synthesize high-quality outputs. Many facial animation works leverage audio Tian et al. (2024); Wang et al. (2025) or landmark-based control Wei et al. (2024); Chen et al. (2024) with diffusion models. While audio offers limited motion guidance, landmarks can be too sparse to express the rich dynamics of facial motion. A recent method, X-Portrait Xie et al. (2024), overcame these problems by using a masked image from the driving video, which contains only the eyes and mouth regions, as the conditional signal to drive the input image. While the results are better compared to previous methods, it struggles significantly with temporal artifacts and large pose variations. Moreover, the model is based on U-Net style architecture (and not the more recent Diffusion Transformer (DiT) architecture), which introduces spatial biases and lacks the capacity to model fine-grained temporal dependencies. Moreover, all the prior methods Guo et al. (2025); Wei et al. (2024); Xie et al. (2024) introduce new modules to incorporate the control signals.

In this paper, we tackle the problem using DiT-based architecture by introducing minimal changes to the base model, while exploiting the power of transformers. This not only helps in faster training, it also generalizes well to various types of input images (see Fig. 4). Moreover, our model can model subtle movements of various face parts, which are perceptually crucial in speech and expression reenactment, thanks to our novel architecture. These results are better appreciated in video format, and we kindly request the reader to check our supplementary video. Our method outperforms all prior methods, establishing a new state-of-the-art.

We follow X-Portrait's Xie et al. (2024) intuition and use masked eyes, nose, and mouth regions of the driving video to control the output video of a given source image. Instead of using the same identity to drive the output during training, which risks appearance leakage, we adopt cross-identity training using the state-of-the-art motion transfer method LivePortrait Guo et al. (2025) to enforce stronger generalization. To effectively integrate control signals, we propose a novel mechanism that reuses the pretrained diffusion model with minimal new parameters. This helps to retain the prior knowledge better while requiring less time to adapt the model to new control signals. Our method also introduces full spatiotemporal attention, allowing each token to interact not only within its frame but also across time. This significantly improves temporal coherence by mitigating flickering artifacts and enables the synthesis of subtle motions. We also provide history frames to the model to provide smooth transitions between consecutive chunks of frames. At inference time, we propose a novel way to combine various control signals that can maintain good identity preservation of the source image while still following motions from the driving video.

In summary, we make the following contributions,

- We propose a pure DiT-based portrait animation model that takes a single source image and a driving video to synthesize high-quality video of the source image following the expressions and poses of the driving video.

- We propose a novel architecture that introduces minimal new parameters and reuses the pretrained diffusion model to incorporate the control signals. This helps in generalizing to

novel identities. Although our model is trained with real human videos, because of prior knowledge from a pretrained diffusion model, our model can generalize to stylized versions of human images. Thanks to our novel architecture, our model outperforms all baselines in capturing subtle lip and facial movements that are perceptually crucial for realistic speech reenactment.

- We provide an extensive qualitative and quantitative evaluation to showcase the advantage of our method over the baselines, with accurate expression transfer and temporally smooth, high-quality output.

## 2 RELATED WORKS

Talking-head generation also referred to as head avatar synthesis, focuses on animating a target face according to motion observed in a driving signal (e.g., another video). Over the past few years, this task has inspired a diverse range of approaches. These can broadly be divided into **GAN-based** and **diffusion-based** methods.

**GAN-based Methods.** A substantial body of work uses pre-defined motion descriptors such as 3D morphable models (3DMM), facial landmarks, or dense flow maps. For example, *FOMM* Siarohin et al. (2020) employs learned keypoints and local affine transformations for animating a source image according to the driving video frames. Many other studies Qian et al. (2024); Xu et al. (2023); Khakhulin et al. (2022) studies incorporate 3D landmarks, blendshapes, or thin-plate splines to better cope with complex head rotations and large expressions.

Instead of explicitly modeling landmarks or 3D structures, several approaches learn latent codes that capture facial and head motion Zakharov et al. (2020); Burkov et al. (2020). *MegaPortraits* Drobyshev et al. (2023) demonstrates the effectiveness of high-resolution, one-shot avatars via latent representations that preserve identity. *EmoPortraits* Drobyshev et al. (2024) focuses on emotional expressiveness, using an expression-rich dataset to achieve more nuanced facial animation. Additionally, *MCNet* Hong & Xu (2023) explores an identity-conditioned memory compensation module to tackle extreme pose changes. *LivePortrait* Guo et al. (2025) extends implicit-keypoint-based video-driven frameworks (e.g. FaceVid2Vid Wang et al. (2021a)) by significantly scaling up the training data, upgrading network architecture, and introducing auxiliary modules for better controllability (e.g., stitching and retargeting), all while maintaining high inference efficiency.

**Diffusion-based Methods.** While GANs have long dominated portrait animation, recent progress in diffusion-based generative models has opened new pathways for high-fidelity synthesis. Early works on diffusion probabilistic models Ho et al. (2020); Song et al. (2021); Rombach et al. (2022) highlighted the potential of iterative denoising in pixel or latent spaces. Since then, evolved pipelines Karras et al. (2022) and alternative formulations Lipman et al. (2023); Liu et al. (2022) have shown improved stability and sampling quality, culminating in state-of-the-art results Esser et al. (2024).

Some diffusion-based approaches integrate explicit control (e.g., keypoints, segmentation masks, or 3D facial priors) into the denoising process Wei et al. (2024); Ostrek & Thies (2024). *AniPortrait* Wei et al. (2024), for instance, injects keypoints into a latent diffusion backbone, preserving coherent facial motion over time. Other methods focus on transferring motion signals directly from driving data. *XPortrait* Xie et al. (2024) avoids explicit landmarks, learning a latent motion representation from cross-identity video pairs; this captures subtle facial expressions yet requires careful training to prevent identity leakage. In contrast, *EchoMimic* Chen et al. (2024) addresses audio-driven synthesis, using a speech-aware temporal module to synchronize lip movements with the spoken content. Despite their impressive generative capabilities, diffusion-based portrait animation still faces challenges such as handling extreme poses and ensuring fully coherent temporal consistency, motivating further research.

## 3 METHOD

Given a single image of a source identity $\mathcal{S}$ and a driving video $\mathcal{D}$ of any identity, the goal is to synthesize a video $\mathcal{T}$ of the source image closely following the expressions and head poses of the driving video. Our model is capable of generating $\mathbf{F}$ frames at a time. To generate longer videos, we use $\mathbf{F'}$ number of previously generated video frames as one of the control signals to generate smooth

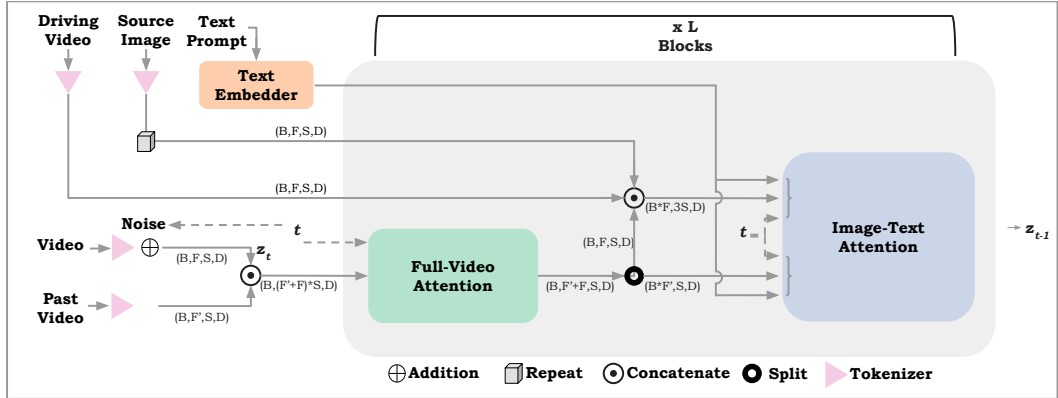

Figure 2: Overview of the method. Our model takes a source image and a driving video as input, and can synthesize a video of the source image following the expressions and poses of the driving video. The model is based on the SD3.5 Medium model. We concatenate the tokens of the source and driving frame to the video noise latents and process them using the Image-Text block. We introduce an additional Full-Video attention block that takes in both video noise latents and tokens of previous video frames to achieve temporally smooth and consistent results with respect to the previous frames.

transitions. Instead of learning the model from scratch, we use a powerful pre-trained diffusion model based on pure transformer architecture as our base and propose novel ways to provide the control signals with minimal changes to the architecture and minimal addition of new parameters. We start with a brief explanation of the diffusion models in Sec 3.1, discuss various control signal and their motivations in Sec 3.2, propose a novel architecture to accommodate the various control signals in Sec 3.3, and finally provide the novel inference strategy in Sec 3.4. We provide a high-level architecture of our model in Fig. 2.

## 3.1 PRELIMINARIES

Diffusion Models (DM)Ho et al. (2020); Song et al. (2021); Rombach et al. (2022) are generative models that learn to map a Gaussian noise distribution to the data distribution through denoising steps. Latent distribution models are a type of diffusion models that apply the same technique in the latent space of the data. This reduces the dimensionality and makes the model efficient. We are interested in image/video-based models for our task. In the literature, mainly 2 types of network architecture have been used to learn this mapping function. The first one is based on U-Net-style architecture, and the second one is based on pure transformer blocks. The objective of noise estimation in a diffusion model is to predict the noise $\epsilon_t$ added at each timestep during the forward diffusion process. The model learns to estimate this noise by minimizing the difference between the true noise and the predicted noise at each timestep. The objective function is:

$$\mathcal{L}_\epsilon = \mathbb{E}_{q(z_0,t)}\left[\|\epsilon_t - \hat{\epsilon}_\theta(\mathbf{z}_t, t)\|^2\right]$$

Where, $\epsilon_t$ is the true noise added at time step $t$, $\hat{\epsilon}_\theta(\mathbf{z}_t, t)$ is the predicted noise by the model parameterized by $\theta$, $\mathbb{E}_{q(z_0,t)}$ denotes the expectation over the clean data distribution $q(z_0)$ and the diffusion process at time $t$. By minimizing this objective, the model learns to reverse the diffusion process effectively, allowing for denoising and generating new samples from noise.

## 3.2 CONTROL SIGNALS

The main control signals for the task are the source image $\mathcal{S}$ and the driving video $\mathcal{D}$. At inference, the identity of $\mathcal{S}$ and $\mathcal{D}$ could be different. To train this model, ideally, one needs paired data of 2 identities performing the same set of expressions and head poses with the same camera position to be able to train in a supervised manner. But practically, it's impossible to obtain such real data. Existing works train the model using video of the same identity to supervise self-driven tasks. That

is, the $\mathcal{S}$ and $\mathcal{D}$ are of the same identity during training. With this setup, there could be leakage of appearance information from $\mathcal{D}$ to the output if not handled carefully. The existing works propose elaborate ways to disentangle appearance from motion information to avoid appearance and identity leakage during self-driven training tasks and can be generalized to novel identities at test time. Our goal is to efficiently use the rich priors of the pre-trained diffusion model and adapt the model for the reenactment task. Following the approach in X-Portrait Xie et al. (2024), we avoid identity leakage by using different identities for $\mathcal{S}$ and $\mathcal{D}$ during training. To ensure consistent motion while varying identity, we employ the state-of-the-art LivePortrait model Guo et al. (2025) to generate paired videos where different subjects perform the same motion patterns. This encourages the model to disentangle motion from appearance. Also, instead of using the complete driving image as input, we only used masked areas of the eyes, nose, and mouth region, which are the most relevant signal for the chosen task.

Our model can synthesize $\mathbf{F}$ frames at a time. To synthesize a longer sequence of video, we need to run the model multiple times in a sliding window manner over the driving video frames. To have smooth transitions between consecutive sets of frames, we also use a set of previous frames for generating the current set of frames. We explain how we make use of these various control signals with minimal changes to the model architecture in the next section.

## 3.3 NETWORK ARCHITECTURE

We use a pure transformer architecture, text-to-image diffusion model as our base model. This model is trained to take in textual data to sample images. At a high level, the network architecture has multiple blocks with two branches, an image branch and a text branch. The image and text tokens interact with each other in a self-attention block with concatenated tokens as input. As it is hard to have all the identity-specific and motion details accurately described in textual space, we mainly resort to image-based conditioning to handle this duty.

*Source Image $\mathcal{S}$*: The identity-specific details come from the source image $\mathcal{S}$. Instead of introducing a new module that can interpret $\mathcal{S}$, we exploit the pure transformer architecture that already has an image branch that can interpret image details well in the form of noise latent $\mathbf{z}$. Specifically, to induce identity information into the noise latent, we simply concatenate the tokens of $\mathcal{S}$ with $\mathbf{z}$. Note, since we generate $\mathbf{F}$ frames at a time, we simply repeat the $\mathcal{S}$ by $\mathbf{F}$ number of times for concatenation with $\mathbf{z}$. To distinguish tokens of $\mathbf{z}$ and $\mathcal{S}$, we simply use different spatial encoding for each of these tokens. Specifically, we shift the width and height of source token positions by a fixed size, which doesn't overlap with that of $\mathbf{z}$ tokens.

*Driving Video $\mathcal{D}$*: The motion information comes from $\mathcal{D}$ and they also have $\mathbf{F}$ frames. As $\mathcal{D}$ is also represented as a set of images, we follow a similar strategy as that of $\mathcal{S}$ and concatenate it with $\mathbf{z}$ and $\mathcal{S}$ in the token dimension, which finally yields $3 \times \mathbf{S}$ tokens for each frame. Similar to $\mathcal{S}$ tokens, to distinguish tokens of $\mathcal{D}$ tokens, we use a different spatial encoding for $\mathcal{D}$ tokens. Note that we haven't introduced any new parameters or major changes to the architecture till now.

*Previous Video $\mathcal{T}'$*: As mentioned before, our model is capable of synthesizing $\mathbf{F}$ frames at a time. To maintain temporal smoothness across consecutive sets of frames, we use previous frames of the target video $\mathcal{T}$ to condition the model. Specifically, we use $\mathbf{F}'$ number of previous frames of target video $\mathcal{T}$ for conditioning, denoted as $\mathcal{T}'$. Since we want to reuse the same network architecture as much as possible, we simply reuse the image-text block to obtain an intermediate representation of $\mathcal{T}'$ and use it in *Temporal Module* to introduce a smooth transition to the current set of output frames.

*Temporal Module*: The base model sd3 is only trained to handle spatial data. We introduce a new module that can handle temporal data. Specifically, we want temporal interaction between frames of $\mathbf{z}$ and $\mathcal{T}'$ to introduce smooth changes over time. Please note, since $\mathcal{D}$ and $\mathcal{S}$ majorly contribute to spatial changes and have a minimal role to play in the temporal aspect, and for efficiency reasons, we exclude tokens corresponding to each of them for temporal modeling. We provide concatenated tokens of $\mathbf{z}$ and $\mathcal{T}'$ in the frame dimension and provide that as input to a full-video attention module. To incorporate the frame number information, we add frame number encoding to both $\mathcal{T}'$ and $\mathbf{z}$.

## 3.4 INFERENCE

Given the presence of multiple control signals during training, we adopt a dropout strategy similar to those used in text-to-image and other conditional diffusion models, where control signals are

randomly dropped during training. This approach enables classifier-free guidance (CFG) at inference time and allows us to modulate the influence of each control signal. Our primary goal is to control the strength of identity details, motion cues, and the influence of previous frames. To achieve this, we run the model with four different input configurations and combine their outputs at each denoising step to produce the final denoised latent representation, $\mathbf{z}$. In the first configuration, all control signals are dropped to enable unconditional generation, denoted as $\mathbf{u}$. In the second, only the source image $\mathcal{S}$ is provided to control identity-specific details, resulting in $\mathbf{s}$. In the third configuration, both $\mathcal{S}$ and the driving video $\mathcal{D}$ are used as control signals, yielding $\mathbf{d}$. Finally, all control signals—including $\mathcal{S}$, $\mathcal{D}$, and previous frames $\mathcal{T}'$—are provided to obtain the fully conditioned output, $\mathbf{p}$.

We combine these 4 outputs in each denoising step in the following way,

$$\mathbf{z} = \mathbf{u} + \lambda_{\mathbf{s}} \times (\mathbf{s} - \mathbf{u}) + \lambda_{\mathbf{d}} \times (\mathbf{d} - \mathbf{s}) + \lambda_{\mathbf{p}} \times (\mathbf{p} - \mathbf{d})$$

## 4 EXPERIMENTS

First, we provide the details of the implementation. Then we present the results and provide a comparison to the baselines considered.

**Implementation Details:** We use Stable Diffusion 3.5 Medium sd3 (SD3.5M) as our base model. SD3.5M is a pure DiT model that has multiple blocks. Each block mainly has 2 branches, one for image and another for text. And, it also has a module that concatenates both image and text outputs to perform self-attention. We introduce a spatio-temporal block before each block SD3.5M, that takes image noise tokens and previous frame tokens and performs full attention where each token in each frame attends to every other token from other frames to obtain spatiotemporal coherence. We train our model in 2 stages. In the first stage, we zero out history frames to avoid the model being overly dependent on history frames. In the second stage, we include history frames for the training. We initialize the model using SD3.5M and fine-tune the whole model on 32 NVIDIA H100 GPUS for around 50k iterations, with a batch size of 1. We use a dataset that was collected internally, which consists of around 20000 clips to train our model. We use a resolution of $576 \times 576$ videos. We set $\lambda_{\mathbf{s}} = 2$, $\lambda_{\mathbf{d}} = 2.5$, $\lambda_{\mathbf{p}} = 1$, $\mathbf{F} = 16$, and $\mathbf{F}' = 3$. Our model takes around 4 seconds to run 1 denoising step. We use 40 steps to get the complete denoised output.

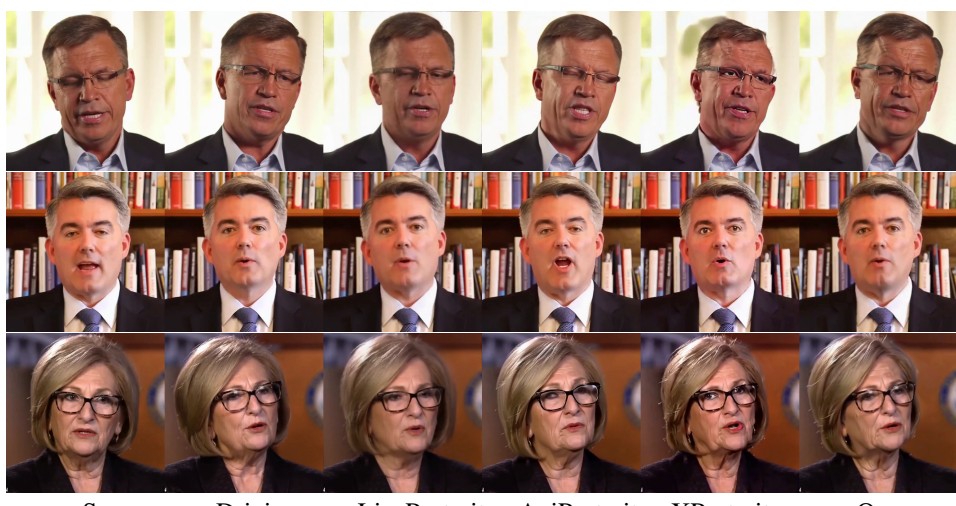

| Source | Driving | LivePortrait | AniPortrait | XPortrait | Ours |

Figure 3: Comparison of Self-Reenactment results on HDTF dataset. Our model outperforms all the other methods in both video quality and accurate expression transfer.

**Baselines:** We compare our method with both non-diffusion-based, such as LivePortrait Guo et al. (2025), and diffusion-based methods, such as XPortrait Xie et al. (2024) and AniPortrait Wei et al. (2024). We use their official implementation to obtain the results.

**Benchmark:** We use HDTF Zhang et al. (2021), TalkingHead1KH Wang et al. (2021b), SD3.5M sd3 model to sample real and different style portrait images, CMU-Mosei Bagher Zadeh et al. (2018)

| Method | Self-Reenactment | | | | | | | |
|---|---|---|---|---|---|---|---|---|
| | $\mathcal{L}_1\downarrow$ | LPIPS↓ | PSNR↑ | FVD↓ | Sync-D↓ | Sync-C↑ | CSIM↑ | MAE↓ |
| LivePortraits Guo et al. (2025) | 0.1084 | 0.1773 | 20.1437 | 82.47 | 7.34 | 7.84 | 0.8808 | 7.49 |
| AniPortrait Wei et al. (2024) | 0.0726 | 0.111 | 22.7084 | 77.85 | 10.09 | 4.83 | 0.8341 | 10.38 |
| X-Portrait Xie et al. (2024) | 0.0811 | 0.1233 | 22.1313 | 74.88 | 8.20 | 6.92 | 0.8581 | 7.63 |
| Ours | **0.0687** | **0.1031** | **22.9669** | **50.31** | **7.31** | **8.01** | **0.9087** | **5.51** |

Table 1: Quantitative comparisons for Self-Reenactment. Our method outperforms all the other baselines across all metrics.

| Method | Cross-Reenactment | | | | |
|---|---|---|---|---|---|
| | FVD ↓ | Sync-D↓ | Sync-C↑ | CSIM ↑ | MAE ↓ |
| LivePortraits Guo et al. (2025) | 174.95 | 8.50 | 6.72 | 0.7811 | 11.02 |
| AniPortrait Wei et al. (2024) | 243.22 | 11.47 | 3.80 | **0.8192** | 18.56 |
| X-Portrait Xie et al. (2024) | 171.70 | 9.32 | 5.90 | 0.7679 | 13.25 |
| Ours | **152.31** | **8.48** | 6.98 | 0.7961 | **10.56** |

Table 2: Cross-Reenactment: Our method outperforms all the other baselines in most metrics. While AniPortrait performs slightly better on identity preservation, it suffers significantly in video quality, eye gaze, and expression transfer metrics.

data for evaluations. To evaluate the performance of test cases, we evaluate the results using a number of metrics. We use L1, LPIPS, and PSNR metrics in case the corresponding image output is available. To evaluate the identity preservation in the output, we use the CSIM metric Deng et al. (2019). To evaluate the expression preservation with respect to driving video, we use the lip synchronization metric (Sync-C, Sync-D) to measure the correlation of lip movement with respect to the audio of the driving signal Chung & Zisserman (2016). Sync-C represents synchronization confidence, and Sync-D represents average synchronization distance. To measure the perceptual quality of video output, we use Content-Debiased FVD Ge et al. (2024). To measure faithful eye movement transfer, we use the Mean Angular Error (MAE) of eye-ball direction Abdelrahman et al. (2023).

### 4.1 SELF-REENACTMENT

We perform self-reenactment on the test set by using the first image as the source image and the rest of the frames as the driving frames of a test video. Specifically, we use HDTF Zhang et al. (2021) to perform the evaluation. We compare both qualitatively and quantitatively to the baselines in the following.

**Qualitative:** We provide a qualitative comparison in Fig. 3. Our method faithfully transfers motion, including both expressions and pose. While LivePortrait does a reasonable job in self-reenactment, it lacks high-frequency details. AniPortrait Wei et al. (2024) relies on facial landmarks as the control signal. While it provides a coarse signal of expressions and pose, landmarks alone aren't sufficient to represent subtle changes in expressions. X-Portrait Xie et al. (2024) works well if the poses are aligned well with the source and driving frame, but suffers significantly with spatial and temporal artifacts otherwise. The reenactment results are better appreciated in the video results. We request the reader to check the supplementary video results.

**Quantitative:** We provide a quantitative comparison in Tab. 1. Our method outperforms all the baselines in all the metrics. While LivePortrait Guo et al. (2025) works well in lip synchronization metrics, it lacks perceptual quality and video quality metrics. AniPortrait Wei et al. (2024) has better perceptual quality, but because of the landmark-based control signal, it suffers significantly in lip synchronization metrics. X-Portrait Xie et al. (2024) performs reasonably, but suffers from spatial and temporal artifacts, which are evident in a drop in the FVD metric.

### 4.2 CROSS-REENACTMENT

We perform cross-reenactment on the source images from TalkingHead-1KH Wang et al. (2021b) and driving videos from HDTF Zhang et al. (2021). To showcase generalization capability, we

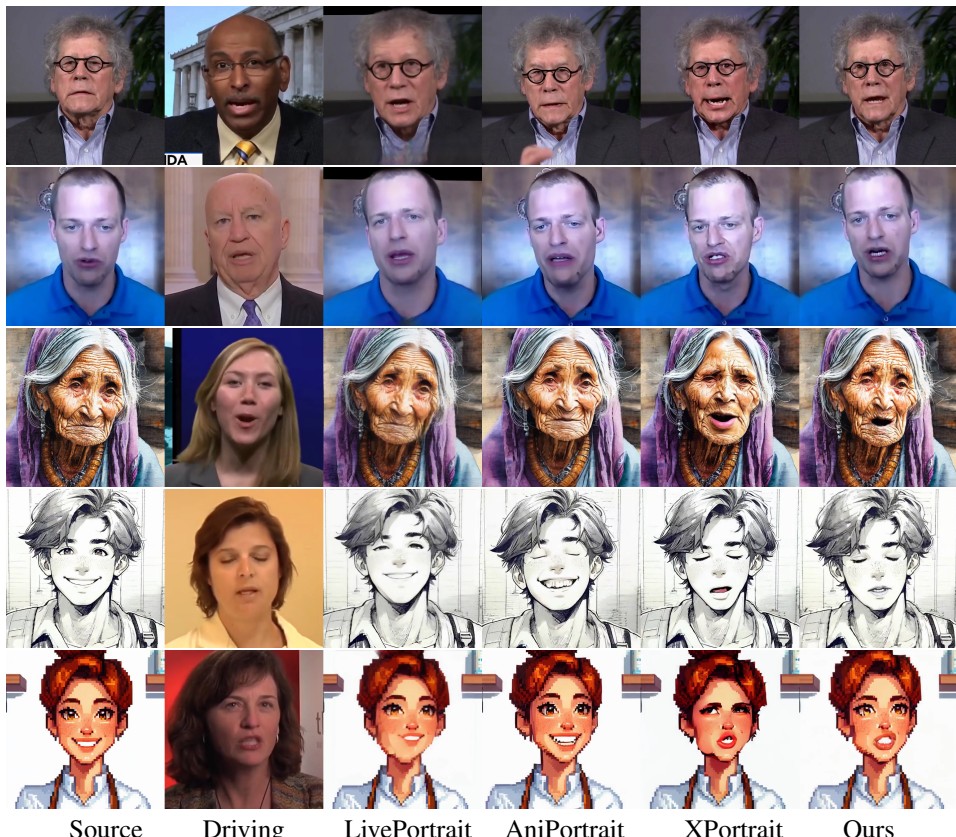

|        |         |              |            |          |      |
| Source | Driving | LivePortrait | AniPortrait | XPortrait | Ours |

Figure 4: Comparison of Cross-Reenactment results using TalkingHead-1KH, HDTF, SD3.5 and CMU-Mosei dataset. In general, LivePortrait fails to generate high-frequency details. AniPortraits fail to transfer expression faithfully, relying on coarse landmarks as the control signal. XPortrait has both spatial and temporal artifacts and has wrong expression predictions. Our model outperforms all the methods in various aspects.

sample images from the Stable Diffusion 3.5 Large sd3 (SD3.5L) model of real and different styles of portraits like sketch, painting, Ghibli, etc, and use the CMU-Mosei dataset Bagher Zadeh et al. (2018) and an internally collected dataset to drive them.

**Qualitative:** We provide a qualitative comparison of cross-reenactment in Fig. 4. Similar to that of self-reenactment, LivePortrait results can not synthesize high-frequency details. AniPortrait only relies on landmarks to control expression; it tends to keep the expression bias of the input source image in the output, ignoring the emotion of the driving input (see rows 2, 3, and 6 in Fig 4). X-Portrait fails to generalize well across different styles. For example, if the source image is of a sketch portrait, X-Portrait fails to keep the output consistent with sketch style (see 4th row in Fig 4, where tongue turns red). The reenactment results are better appreciated in the video results. We request the reader to check the supplementary video results.

**Quantitative:** We provide the quantitative comparison in Tab. 2. Our methods outperform all the baselines in most of the metrics. AniPortrait Wei et al. (2024) performs slightly better in the identity metric(CSIM). This, we believe, is because this baseline retains the expression bias of the source image in the output and, as a result, could have influenced the identity metric. This impacts the quality of expression transfer, which is evident in its poor performance in the lip synchronization metric. Since the landmark doesn't represent the eye gaze, their eye direction metric (MAE) is quite bad as well. LivePortrait fails to generalize well to stylized images (see 4th row in Fig. 4). XPortrait struggles when the driving pose is different from that of the source image (see 5th row in Fig. 4)

### 4.3 ABLATION STUDY

We provide 2 ablation studies for our model.

**Factorized Attention (FA):** To model the temporal aspect, we use full video attention where each token of a frame attends to all the tokens of all the frames. We provide an ablation study where we replace full video attention with only factorized attention, where a token in each frame attends to only corresponding tokens in other frames. The quantitative comparison can be found in Tab. 3. We observe that although the model with factorized attention has slightly better identity and video quality metrics, it struggles to have better lip synchronization. We choose the model with full video attention as our main model, which is used for generating all the results shown.

| Method | FVD ↓ | Sync-D↓ | Sync-C↑ | CSIM ↑ | MAE ↓ |
|---|---|---|---|---|---|
| Ours (w/o history) | 202.29 | 8.72 | 6.72 | 0.7833 | 11.13 |
| Ours | **152.31** | **8.48** | **6.98** | **0.7961** | **10.56** |

Figure 5: Training curriculum ablation. Observe the accumulated error patches on the forehead for the model without careful curriculum (CC).

Table 3: Factorized Attention ablation. Although the model with factorized attention has better identity and video quality metrics, it has poor performance in lip synchronization and eyeball movement transfer.

**Careful Curriculum (CC):** The training curriculum determines the quality of the model. Although the previous frames signal helps in providing smooth transitions in the output, training the model with that signal for the entire training makes the model overly rely on the history frames. When generating a longer sequence of output, we need to run the model multiple times by using the previous run's output. If the output has a minor error, overreliance on the previous frames results in the accumulation of error over time. To avoid this, we pretrain our model by zeroing out the previous frames' input and fine-tune with this signal for the last 10k iterations. We provide the comparison in Fig 5. One can observe the accumulation of error resulting in the dark artifacts on the forehead for the model trained without the careful curriculum.

## 5 LIMITATIONS

Although our method works well on most of the real-human portraits and their stylized versions, like sketch, painting, pixart, etc., it doesn't work well on extreme cases where the proportions of face parts are not similar to those of real human faces. One such failure sample is shown in Fig. 6. We can observe that the source image has eyes that are close to the nose and the mouth. Our model mistakes the eyebrows for the eyes, resulting in the wrong output.

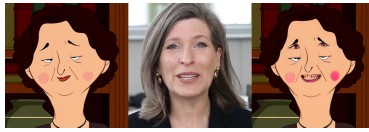

Source    Driving    Output

Figure 6: Limitation.

## 6 CONCLUSION

In this work, we presented a diffusion transformer-based approach for high-quality portrait animation using a single source image and a driving video. Our method addresses key challenges in existing video-driven reenactment systems, including temporal inconsistency, identity leakage, and limited generalization to diverse appearances. By leveraging masked facial regions as expressive control signals, adopting cross-identity training via a motion transfer model, and introducing full spatio-temporal attention mechanisms, our model achieves accurate and temporally coherent outputs. Our model has better lip synchronization than the state-of-the-art methods. Furthermore, our strategy for integrating control signals into a pretrained diffusion transformer requires minimal additional parameters and enables strong generalization, even to stylized human-like inputs. Extensive experiments demonstrate the superiority of our approach over prior methods, both qualitatively and quantitatively.

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

SUPPLEMENTARY

1. **Video Results**

   We present comparative video results in the supplementary video, including self-reenactment and cross-reenactment on both real and stylized human subjects. Our method consistently outperforms all the baseline approaches. These improvements are most evident in motion and visual quality, and we strongly encourage readers to view the supplementary video to fully appreciate the results.

2. **Implementation Details**

   Here we provide more implementation details. We used $0.0001$ as our learning rate with a batch size of 1. We drop out $\mathcal{S}$, $\mathcal{D}$, and $\mathcal{T}'$ at $50\%$ of iterations each during training. Our network architecture is based on SD3.5 Medium model, where we introduce a full video attention module in all the blocks of SD3.5 Medium model. We optimize all the parameters of the model during training.

3. **Broader impact** In the entertainment and media industry, it enables more immersive and realistic visual effects, bringing historical figures to life or allowing actors' performances to be altered without requiring reshoots. In accessibility, it offers promising tools for generating expressive avatars for people with disabilities. However, the technology also raises serious ethical and societal concerns, especially in the context of misinformation and deepfakes. The ability to convincingly alter facial expressions can be exploited to fabricate videos for malicious purposes, potentially undermining public trust in digital media and enabling identity fraud. As face reenactment technology continues to advance, its broader impact underscores the urgent need for responsible development, regulation, and public awareness to ensure it is used for beneficial, rather than harmful, applications.

4. **Societal Impact**

   The societal impact of face reenactment technology is significant and complex, as it challenges traditional notions of authenticity and trust in visual media. By enabling the realistic manipulation of facial expressions and identities in videos, face reenactment can blur the line between genuine and fabricated content. This has profound implications for public discourse, journalism, and personal privacy. On one hand, the technology can be used for creative expression, education, and accessibility, but on the other, it poses serious risks when used to create deepfakes for political manipulation, defamation, or cyberbullying. The widespread availability of such tools can erode trust in video evidence, making it harder to distinguish truth from deception in an already polarized information environment. As a result, face reenactment not only raises technical and ethical challenges but also demands urgent societal engagement to develop safeguards, promote media literacy, and establish legal and regulatory frameworks to prevent misuse.

