# OpenReview forum: "Stable Video-Driven Portraits"
_ICLR.cc/2026/Conference — Submitted to ICLR 2026_

### Official Review · Reviewer_rTAL · 2025-10-24

**Soundness:** 2
**Presentation:** 2
**Contribution:** 2
**Rating:** 2
**Confidence:** 4

**Summary:**

This paper proposes Stable Video-Driven Portraits, a video-driven portrait generation framework based on the Diffusion Transformer (DiT) architecture. The model takes a single source image and a driving video as input and synthesizes high-fidelity dynamic videos that reenact the target person’s motion and expressions.
The main contributions claimed are as follows:
1. Adoption of a cross-identity training scheme combined with facial-region masking (eyes, nose, mouth) to prevent identity leakage;
2. Minimal-parameter adaptation on top of Stable Diffusion 3.5, achieving faster convergence and improved generalization;
3. Introduction of a full spatio-temporal attention mechanism to capture fine-grained motion and enhance temporal coherence.

**Strengths:**

1. The Supplementary Material includes detailed technical explanations and training pipeline descriptions, which improve readability and aid understanding.
2. The experiments are comprehensive, covering self-reenactment, cross-reenactment, and stylized portrait animation.
3. The evaluation metrics are diverse, including PSNR, FVD, CSIM, and MAE, allowing a thorough quantitative evaluation.
4. The method effectively leverages the pretrained SD3.5 model with minimal parameter adjustments, demonstrating practical adaptation of large diffusion transformers.
5. The cross-identity training strategy effectively prevents appearance leakage and improves generalization.
6. The full spatio-temporal attention mechanism significantly enhances temporal smoothness and consistency in generated videos.
7. The ablation studies verify the rationale of the training curriculum and attention structure.
8. The paper includes comparisons with non-Diffusion Transformer architectures, validating the advantages of the proposed DiT-based design.

**Weaknesses:**

1. Writing and organizational quality are significantly below ICLR standards. The paper suffers from vague phrasing, redundant sections, missing implementation details, and low-quality figure presentation. For example, the Broader Impact and Societal Impact sections in the supplementary material repeat nearly identical content, showing a lack of editorial care. The overall narrative feels fragmented and lacks the polished academic tone typically found in ICLR-accepted papers.
2. The abstract and introduction are overly generic and imprecise. Keywords such as “minimal new parameters” and “superior temporal consistency” are not supported with specific numbers or quantitative evidence. For instance, it is unclear exactly how many additional parameters were introduced or what percentage of the base SD3.5 model this represents. These missing details weaken the clarity and perceived magnitude of the contribution.
3. Limited novelty. The core technical design heavily depends on Stable Diffusion 3.5 with only minor adaptations. The use of masked facial regions (eyes, nose, mouth) as input features is already a well-established practice in talking-head generation literature. Consequently, the originality of the approach appears limited and largely incremental.
4. Method–result mismatch. The paper states that only masked facial regions are used for motion input, which theoretically should exclude head pose changes. However, the supplementary video results clearly show head movements, suggesting inconsistencies between the described methodology and the observed behavior.
5. Excessive reliance on LivePortrait. The training pairs are generated using LivePortrait, which likely introduces error accumulation. Moreover, the data generation process using LivePortrait is not well-documented. It is also unclear why a model trained on LivePortrait-generated data produces results superior to LivePortrait itself, and no ablation or justification is provided for this discrepancy.
6. Insufficient comparison with other methods. The experiments compare against only three baselines, which is inadequate for a top-tier conference. Stronger and more recent competitors such as VividPortraits, DiffPortrait, and AniFace should be included to strengthen the argument.
7. Implementation details are lacking. Critical aspects such as token dimensionality, dropout scheduling, temporal encoding design, and optimizer settings are omitted, making reproducibility difficult.
8. Ablation studies are incomplete. The paper only provides ablations on attention factorization and training curriculum. Important factors such as region-wise mask contributions (eyes vs. mouth vs. nose), sensitivity to fusion weight λ, and additional parameter count are missing. Including these would significantly improve the paper’s technical credibility.
9. Internal dataset not released. The claimed 20k internal dataset is neither released nor described in sufficient detail (e.g., domain diversity, subject identity distribution, video length), preventing reproducibility and verification by the research community.

**Questions:**

1. Since LivePortrait-generated paired data are used for training, how does the proposed model outperform LivePortrait itself?
2. Given that only masked facial regions (eyes, nose, mouth) are used as motion input, why do the generated results still include head pose movements?
3. In classifier-free guidance, all control signals—including the source image—are discarded in one configuration. Does this impact identity preservation?
4. Do the authors plan to release the internal 20k video dataset? If not, please describe its distribution, diversity, and sampling details.
5. Could the proposed framework be extended to audio-driven or emotion-conditioned portrait generation?
6. Have the authors evaluated longer sequences (e.g., >16 frames) or longer reference clips (>3 frames) for temporal consistency analysis?
7. Does the full-video attention scale quadratically with frame length, and how is inference efficiency maintained in such cases?

---

### Official Review · Reviewer_wD8U · 2025-10-25

**Soundness:** 2
**Presentation:** 2
**Contribution:** 2
**Rating:** 2
**Confidence:** 4

**Summary:**

This paper proposes a diffusion transformer (DiT)-based framework for high-quality video-driven portrait animation, which synthesizes a video of a source identity (from a single image) that faithfully replicates the expressions and poses of a driving video. Addressing key limitations of existing methods—temporal inconsistency, identity leakage, and poor generalization to stylized inputs—the framework leverages three core innovations: Masked Control Signals, Cross-Identity Training and Spatio-Temporal Attention & Inference Fusion. Evaluated on public datasets (HDTF, TalkingHead-1KH, CMU-Mosei) and stylized inputs (sketches, Ghibli-style), the method outperforms baselines (LivePortrait, AniPortrait, X-Portrait) across metrics.

**Strengths:**

- Unlike U-Net-based diffusion models (e.g., X-Portrait, Xie et al. 2024), which introduce spatial biases and struggle with temporal dependencies, the proposed DiT-based design (built on SD3.5 Medium) enables flexible token-level interactions. This allows the model to capture subtle motions (e.g., lip movements during speech) that are critical for realism—quantitatively reflected in superior lip synchronization (Sync-D: 7.31 vs. 8.20 for X-Portrait) and eye gaze transfer (MAE: 5.51 vs. 7.63 for X-Portrait).
- Most diffusion-based methods (e.g., AniPortrait, Wei et al. 2024) add large task-specific modules to integrate control signals, increasing training time and overfitting risk. In contrast, this framework reuses the pre-trained SD3.5 backbone with minimal new parameters (only spatio-temporal attention blocks), retaining rich generative priors and enabling faster convergence. This also explains its strong generalization to stylized inputs (e.g., sketches), which benefit from SD3.5’s pre-trained knowledge of diverse visual styles.
- GAN-based methods (e.g., LivePortrait, Guo et al. 2025) and self-driven diffusion models often suffer from identity leakage (transferring driving identity features to the source). The paper’s cross-identity training (using LivePortrait to generate paired cross-identity motions) explicitly disentangles motion from appearance, leading to better identity preservation (CSIM: 0.7961 vs. 0.7811 for LivePortrait in cross-reenactment) while maintaining motion fidelity.
- Baselines like X-Portrait rely on frame-wise processing, causing flickering artifacts. The proposed full spatio-temporal attention (each token attends to all tokens across frames) outperforms factorized attention (token-wise cross-frame attention) in temporal coherence—evident in lower FVD (152.31 vs. 202.29 for the no-history variant) and reduced error accumulation in long videos (via a "careful curriculum" training strategy that avoids over-reliance on history frames).

**Weaknesses:**

- The model is trained on an internal dataset of 20,000 clips, with no details on identity diversity, lighting conditions, or pose variations—hindering reproducibility. In contrast, SOTA works like Durian (from prior reviews) use public datasets (CelebV-Text, VFHQ) for transparency. Additionally, the paper only compares to three baselines, omitting recent diffusion-based methods (e.g., StrucBooth, which optimizes fine-grained structural details) that could contextualize its performance on expression fidelity.
- Training requires 32 NVIDIA H100 GPUs for 50,000 iterations—far beyond the reach of most research teams. This is in stark contrast to lightweight methods like StrucBooth (trained on a single 48G A6000 GPU) or LoRA-based fine-tuning approaches. The paper also does not explore optimizations (e.g., parameter-efficient fine-tuning, model distillation) to reduce resource demands.
- Inference takes 160 seconds per video (4 seconds per denoising step × 40 steps)—a critical limitation for real-world applications (e.g., virtual meetings, live streaming). Baselines like LivePortrait achieve real-time inference (~30 FPS), while even diffusion-based methods like StrucBooth require only 800 optimization steps for similar quality.
- The model struggles with non-human-like facial geometries (e.g., source images with eyes abnormally close to the nose/mouth), mistaking features (e.g., eyebrows for eyes) and producing invalid outputs. This contrasts with 3D-aware methods like MegaPortraits (Drobyshev et al. 2023), which handle extreme proportions via mesh-based avatars.
- The paper does not evaluate performance when the driving video contains occlusions (e.g., hands covering the mouth, glasses reflections). Existing methods like EchoMimic (Chen et al. 2024) use landmark editing to mitigate occlusions, but this framework lacks such safeguards—likely leading to motion artifacts in occluded scenarios.
- While the model works for sketches and Ghibli-style inputs, it is untested on non-human stylized figures (e.g., cartoon characters with exaggerated features like oversized eyes or no noses). This limits its utility for creative applications (e.g., animated film production) where diverse stylizations are required.

**Questions:**

- What is the composition of the internal 20,000-clip dataset (e.g., identity diversity, pose/expression range, lighting variations)? Will the dataset be made public to enable reproducibility?
- Can parameter-efficient techniques (e.g., LoRA) or reduced denoising steps (e.g., 20 instead of 40) maintain performance while lowering training/inference costs?
- How would the model perform if the driving video includes heavy occlusions or extreme poses (e.g., 90° head turns)? Could integrating 3D facial priors (e.g., 3DMM) improve robustness?

---

### Official Review · Reviewer_Gnvm · 2025-11-01

**Soundness:** 2
**Presentation:** 2
**Contribution:** 2
**Rating:** 4
**Confidence:** 4

**Summary:**

The paper studies video-driven portrait animation, aiming to generate temporally coherent videos that reenact a source identity from a driving video. It proposes Full-Video Attention for spatiotemporal interaction and History-frame conditioning for continuity. These designs improve temporal smoothness and identity preservation while efficiently adapting an image diffusion backbone (SD3.5) to video. The model produces visually coherent videos with realistic facial motion, showing strong qualitative results compared to prior methods.

**Strengths:**

1. The paper proposes an elegant adaptation of an image diffusion backbone (SD3.5) to a video reenactment task using minimal additional parameters. This design demonstrates parameter efficiency and training stability.
2. The generated videos exhibit clear identity retention and stylistic fidelity, benefiting from SD3.5’s strong image priors. Visual comparisons indicate improved fine-grained motion transfer and facial realism.
3.  The combination of full-video attention and history conditioning is intuitively sound and leads to visibly smoother transitions between segments.

**Weaknesses:**

1. While the paper introduces full-video attention and history-frame conditioning, temporal smoothness is evaluated only indirectly through the **FVD** metric. No explicit temporal-consistency measures (e.g., **tOF**, **tLPIPS**) are reported, and there is no comparison with true video diffusion backbones (e.g., Wan2.1). It remains unclear to what extent the proposed method approaches the temporal coherence achievable by video foundation models.

2. The model uses masked eye–nose–mouth regions from the driving video as motion cues, yet these masks are prone to temporal jitter due to landmark noise, pose variation, or occlusion. The paper does not include robustness or ablation study under imperfect or noisy masks, which is critical for real-world deployment.

3. The method is built upon an image diffusion model, without exploring fine-tuning from available video foundation models that already encode long-term motion priors. Since such models (e.g., Wan2.1) are publicly available, adopting them could provide stronger temporal consistency with little additional cost. The decision to stay with an image-only base seems conservative and possibly leaves performance unrealized.

**Questions:**

1. Please provide explicit temporal-consistency metrics (e.g., tOF, tLPIPS) to directly measure frame-to-frame smoothness.

2. A side-by-side comparison with a recent video diffusion baseline would clarify whether the proposed architecture achieves competitive coherence and motion fidelity.

3. Evaluate stability under synthetic perturbations (e.g., random mask jitter, partial occlusion, dropout).

---

### Official Review · Reviewer_Nrdd · 2025-11-01

**Soundness:** 2
**Presentation:** 2
**Contribution:** 2
**Rating:** 2
**Confidence:** 4

**Summary:**

The paper presents a diffusion-based framework for portrait animation that generates realistic talking-head videos from a single source image and a driving video. Unlike earlier warping or landmark-based methods that struggle with temporal stability and subtle motion, the proposed approach leverages the powerful spatial-temporal reasoning of Diffusion Transformers (DiT) to achieve stable, identity-preserving reenactment. The model reuses the pretrained DiT backbone with minimal new parameters, maintaining pretrained priors while allowing conditional control. A full-video attention mechanism captures inter- and intra-frame dependencies, and history frames ensure continuity across temporal chunks. During inference, a multi-configuration CFG-style fusion balances motion fidelity with identity consistency.

**Strengths:**

This paper presents a clean and stable implementation of video-driven portrait generation built on the powerful SD3.5, demonstrating that large pretrained T2I diffusion backbones can be effectively adapted for temporal control with minimal modification, showing clear improvements in visual fidelity, temporal consistency, and expression synchronization compared to prior U-Net–based methods such as X-Portrait and LivePortrait.

**Weaknesses:**

1. The method is largely an incremental upgrade of X-Portrait—essentially porting the cross-id driving signal and masked-region conditioning to the SD3.5 Diffusion Transformer backbone.
2. The method section is notably under-detailed and heavily relies on a single overview figure to convey the overall architecture.
3. The paper’s cross-identity supervision fully depends on LivePortrait-generated videos to form training pairs. However, if LivePortrait fails, or is insensitive to certain extreme or rare motions, this will prevente the DiT from learning to handle such challenging dynamics.
4. The experimental comparison is unbalanced and somewhat outdated. Most baselines (LivePortrait, AniPortrait, X-Portrait) are based on U-Net or landmark-driven architectures, while the proposed method uses a much stronger SD3.5 DiT backbone.

**Questions:**

Please check the weakness part.

---

### Meta-Review · Area_Chair_iuFA · 2026-01-06

**Summary:**

This paper proposes a DIT-based framework for video-driven portrait animation, adapting Stable Diffusion 3.5 to generate temporally coherent talking-head videos from a single source image and a driving video. Reviewers mostly acknowledge the strong visual quality, the capability of identity preservation and temporal smoothness, but raise substantial concerns that informed the overall negative recommendation, regarding novelty, reproducibility, evaluation rigor, and practical feasibility. The main concerns considered by AC include:

- **Limited technical novelty.** Method is largely an incremental adaptation of X-Portrait, porting existing techniques (masked control, cross-identity supervision) to the SD3.5 DiT backbone.
- **Reproducibility issues.** Undisclosed internal dataset (20k clips), missing implementation details, and insufficient documentation of LivePortrait-based data generation.
- **Inadequate evaluation.** Only three U-Net-based baselines are compared, omitting recent competitors (VividPortraits, DiffPortrait) and modern video diffusion models (Wan2.1). Missing explicit temporal consistency metrics (tOF, tLPIPS).
- **Methodological concerns.** Unexplained reliance on LivePortrait for training pairs with unclear error accumulation. Inconsistency between stated methodology (masked facial regions only) and observed results (head pose movements). Incomplete ablations on mask contributions and fusion weights.
- **Poor presentation.** Writing with vague, several unsupported claims and low-quality figures.

Overall, while the paper demonstrates strong engineering and visually compelling results, reviewers consistently judge the contribution to be incremental, insufficiently validated, and below the bar for ICLR in its current form.

**Reviewer Concerns:**

As no rebuttal has been posted, the concerns remain, as listed below.

Reviewer Nrdd (score: 2) emphasizes that the method is largely an incremental upgrade over X-Portrait, primarily porting existing masked control and cross-identity supervision to a significantly stronger SD3.5 DiT backbone. Reviewer Nrdd also highlights insufficient methodological detail, heavy reliance on LivePortrait-generated supervision, and unbalanced comparisons against weaker U-Net-based baselines.

Reviewer Gnvm (score: 4) raised concerns about the lack of explicit temporal consistency metrics, the absence of comparisons with modern video diffusion models, and the missing robustness analysis under noisy or imperfect masks.

Reviewer wD8U (score: 2) raised major concerns regarding reproducibility (undisclosed internal 20k dataset), extremely high training and inference costs, limited baseline coverage, poor robustness to occlusions and atypical facial geometries, and missing evaluations on non-human stylized portraits.

Reviewer rTAL (score: 2) provided a detailed critique, noting limited technical novelty, poor writing and organization, missing implementation details, insufficient ablation studies, unexplained dependence on LivePortrait for supervision, and inconsistencies between the described methodology and observed results. Reviewer rTAL also highlighted the lack of strong contemporary baselines and the absence of dataset release or detailed characterization.

**Reviewer Scores:**

As there is no rebuttal/discussion provided, the reviewers are supposed to maintain their original scores.

---

### Decision · Program_Chairs · 2026-01-26

Reject